# Screening potential anti-osteoarthritis compounds using molecular docking based on MAPK and NFκB pathways and validating their anti-osteoarthritis effect

Tian-Wang Zhu[1], Yu Zheng[2], Rui-Xin Li[1]*

1 Department of Sports Medicine, Dalian University Affiliated Xinhua Hospital, Dalian, Liaoning, China,
2 Department of Spinal Surgery, Dalian University Affiliated Xinhua Hospital, Dalian, Liaoning, China

* m-cc@163.com

## Abstract

Osteoarthritis is an extremely common disease. However, it lacks effective nonsurgical treatments. Molecular docking has been widely used in drug discovery. However, no studies focus on screening anti-osteoarthritis compounds using molecular docking. This study aimed to screen potential anti-osteoarthritis compounds using molecular docking and validate their anti-osteoarthritis effect. Molecular dockings between 51 compounds inhibiting the MAPK and NFκB pathways but have not been used to treat osteoarthritis and 5 core human proteins in the MAPK and NFκB pathways were performed. Corilagin, Apigetrin, Protopine, 5-methoxyflavone, and 7,3',4'-trihydroxyisoflavone were selected. The drug-likeness, pharmacokinetics, bioactivity, and toxicity of the selected compounds were analyzed. The cytotoxicity and anti-osteoarthritis effect of the selected compounds were tested on mouse chondrocytes. This study found that molecular docking based on the MAPK and NFκB pathways can be used to screen potential anti-osteoarthritis compounds, providing a perspective on drug discovery through pathway-based screening. ERK2, JNK2, and p38 showed similar binding sites commonly interacting with the compounds. The theoretical drug-likeness, pharmacokinetics, bioactivity, and toxicity were largely consistent with the empirical cytotoxicity and anti-osteoarthritis effect. Additionally, Protopine, 5-methoxyflavone, and 7,3',4'-trihydroxyisoflavone showed strong anti-osteoarthritis potential and can be considered for future studies to test their anti-osteoarthritis effect in animal models, explore molecular mechanisms, and improve their solubility.

## Introduction

Osteoarthritis (OA) is an extremely common disease, with 527.81 million cases reported worldwide in 2019 [1]. OA is a leading cause of joint pain and a major cause of disability, and exerts a significant humanistic and economic burden on patients [1,2]. Currently, joint replacement is considered the only way to correct deformity, eliminate pain, and restore functionality in daily activities [3]. However, the trauma and cost associated with joint replacement are significant. Therefore, exploring nonsurgical treatments for OA is crucial.

Data availability statement: All relevant data are within the manuscript and its Supporting Information files.

Funding: R.X. Li received funding from the Dalian Key Medical Specialties "Peak Climbing Program" of the People's Government of Dalian Municipality (https://www.dl.gov.cn/). The funder played no role in the study design, data

collection and analysis, decision to publish, or preparation of the manuscript.

**Competing interests:** The authors have declared that no competing interests exist.

The main pathological feature of the onset and progression of OA is the degradation of chondrocytes and the disintegration of the extracellular matrix. The OA chondrocytes produce excessive nitric oxide (NO), causing continuous release of various inflammatory factors [4]. Interleukin (IL)-1β, an inflammatory factor, is widely considered to participate in the pathways mediating the degradation of chondrocytes and disintegration of the extra-cellular matrix, such as the mitogen-activated protein kinase (MAPK) and nuclear factor-κB (NFκB) pathways [5,6]. Many studies found that activation of the MAPK and NFκB pathways promotes the apoptosis and inflammation of chondrocytes[6–8]. The core proteins in the MAPK pathway are extracellular regulated protein kinases 2 (ERK2), c-Jun N-terminal kinase 2 (JNK2), and p38 [6,9]. The core proteins in the NFκB pathway are p65 and NFκB inhibitor α (IκBα) [7,9]. Because the activation of ERK2, JNK2, p38, and p65 induces OA while IκBα inhibits p65 from transporting into the nucleus, compounds competitively binding to ERK2, JNK2, p38, and p65 but not IκBα may have potential in treating OA (Fig 1).

## Materials and methods

### Reagents

Corilagin (C79750), Apigetrin (A25650), Protopine (P41851), 5-methoxyflavone (Y30875), 7,3',4'-trihydroxyisoflavone (T17490), Cell Counting Kit-8 (AC10873), and NO detection kit (AC10317) were from Shanghai Acmec Biochemical Co., Ltd., Shanghai, China. Type II collagenase (17101015) and trypsin (1734858) were from Thermo Fisher Scientific Inc., Waltham, MA, USA. 4',6-diamidino-2-phenylindole (DAPI) (C0060), Triton X-100 (T8200),

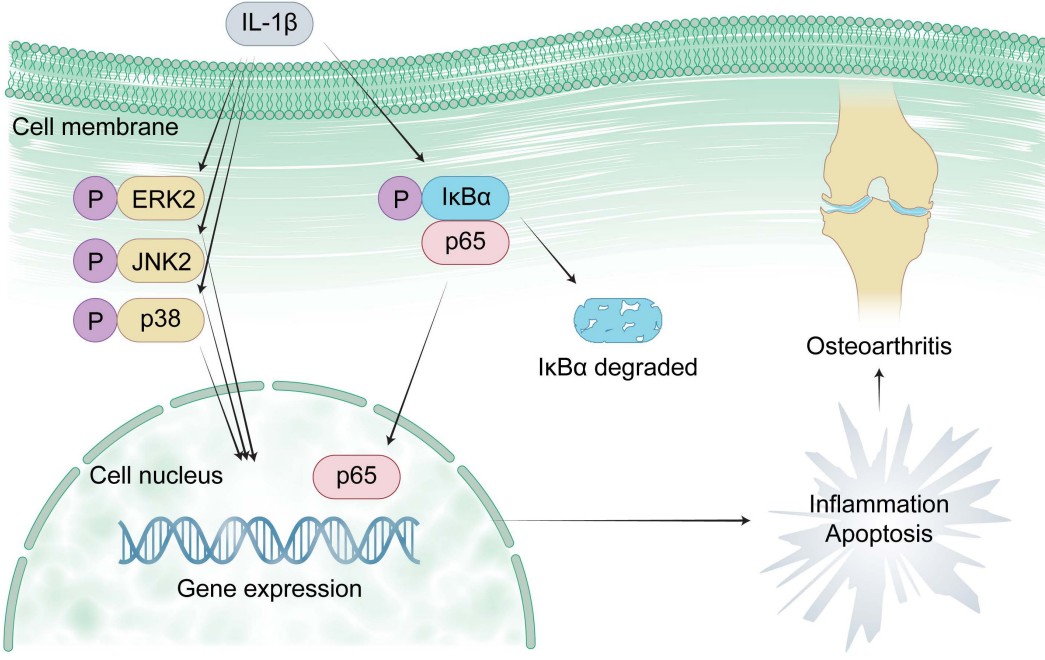

**Fig 1. The roles of the MAPK and NF κB pathways in the onset and progression of osteoarthritis.** Molecular docking is a powerful computational technique used to simulate the interaction between a compound and a protein to evaluate their binding potential. As a strong competitor to high-throughput omics, this technique has been widely used in drug discovery [10]. However, to our knowledge, no studies focused on screening anti-osteoarthritis compounds using molecular docking. This study aimed to screen potential anti-osteoarthritis compounds by docking with core human proteins in the MAPK and NFκB pathways, analyze their drug-likeness, pharmacokinetics, bioactivity, and toxicity, and test their cytotoxicity and anti-osteoarthritis effect on mouse chondrocytes.

and goat serum (S9070) were from Solarbio Science & Technology Co., Ltd., Beijing, China. Type II collagen (15943-1-AP) and Alexa Fluor 488 labeled goat anti-rabbit lgG (SA00006-2) were from Proteintech Group, Inc., Chicago, IL, USA. Active recombinant mouse IL-1 beta protein (RP01340) was from ABclonal Technology Co., Ltd., Wuhan, Hubei, China. Complete chondrocyte culture medium was from Wanwu Biotechnology Co., Ltd., Hefei, Anhui, China. Dulbecco's Modified Eagle Medium: Nutrient Mixture F-12 (DMEM/F12) complete culture medium (PM150312B) was from Pricella Life Science & Technology Co., Ltd., Wuhan, Hubei, China.

## Study design

This study was approved by the Ethics Committee of Dalian University Affiliated Xinhua Hospital (approval number: 2023-55-01). The study design is shown in Fig 2.

## Selection of proteins and compounds for molecular docking

Because the MAPK and NFκB pathways, two classical pathogenic signaling pathways, play important roles in the onset and progression of OA, the core proteins in MAPK and NFκB pathways are considered potential therapeutic targets for OA [8]. Therefore, 6 core proteins in the MAPK and NFκB pathways were selected for molecular docking. They were selected from the pathways provided by BATMAN-TCM (http://bionet.ncpsb.org.cn/

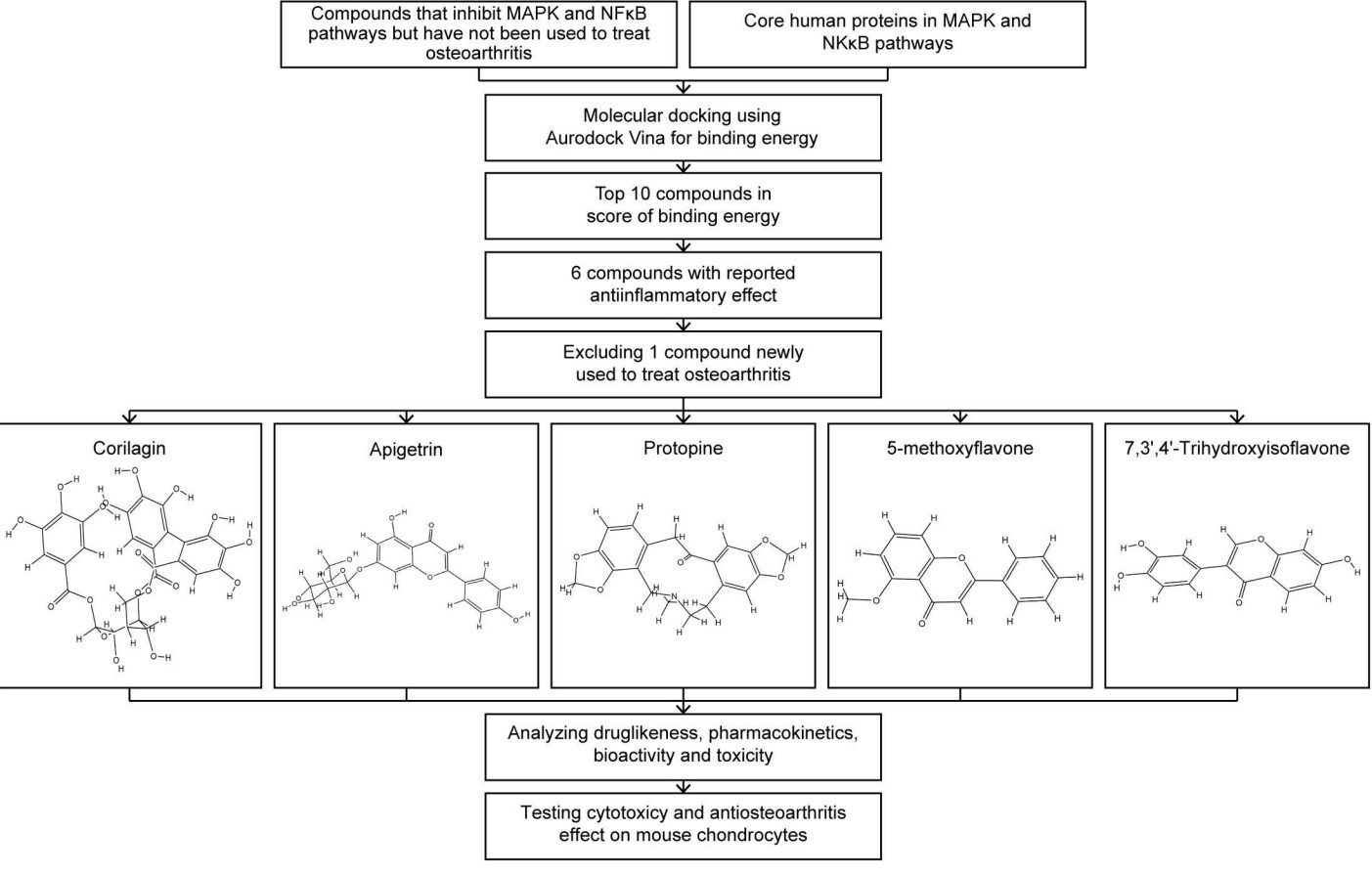

**Fig 2. Study design.**

batman-tcm/index.php/Home/Ontology/index/)[11,12]. The structures of ERK2, JNK2, p38, and p65 were selected from the Protein Data Bank (PDB) (https://www.rcsb.org/) [13]. Because the matched structure of IκBα was not available in PDB, it was selected from the structures provided by AlphaFold Protein Structure Database (AFDB) (https://alpha-fold.ebi.ac.uk/) [14]. The scientific name of the source organism was *Homo sapiens*. The structures were selected based on the degree of match and resolution. The proteins used for molecular docking were ERK2 (8AOJ), JNK2 (7N8T), p38 (6QDZ), p65 (1NFI), and IκBα (AF-P25963-F1-v4).

Because the MAPK and NFκB pathways are highly conserved across different cells, compounds inhibiting the MAPK and NFκB pathways but have not been used to treat OA may still inhibit MAPK and NFκB pathways in chondrocytes [15,16]. Therefore, 51 compounds inhibiting MAPK and NFκB pathways but have not been used to treat OA were selected for molecular docking. They were selected based on a literature review using METSTR (https://www.metstr.com/) between 2021 and 2023. The search term used was "((MAPK[TI] AND NF-κB[TI]) NOT osteoarthritis[TI]) AND 2021:2023[dp]". The compounds used for molecular docking were Rutaecarpine (65752), Sesamolin (101746), Corilagin (73568), Apigetrin (5280704), Gamabufotalin (259803), Protopine (4970), 5-methoxyflavone (94525), 7,3',4'-trihydroxyisoflavone (5284648), Viaminate (6438483), KAN0438757 (71586631), Savinin (5281867), Maraviroc (3002977), 6-methoxyflavone (147157), Sargachromenol (10455044), Tryptanthrin (73549), Chryseriol (5280666), Mogrol (14525327), Nuciferine (10146), Dehydromiltirone (3082765), Rosmanol (13966122), W54011 (5311122), Oxocrebanine (3084713), Chrysoeriol (5280666), Matairesinol (119205), Fluorofenidone (11851183), Demethoxycurcumin (5469424), Eupatilin (5273755), Xanthenone (7020), Tetrahydropalmatine (72301), Harmine (5280953), Isosinensetin (632135), Xanthotoxol (65090), Cl-Amidine (24970878), 3,3',4,5'-tetramethoxy-trans-stilbene (5387294), Pteryxin (5281425), 6-methylcoumarin (7092), Ibudilast (3671), Fingolimod (107970), Xanthatin (5281511), Guanosine (135398635), Nardostachin (196699), Dehydrozingerone (5354238), Scopoletin (5280460), Bicyclol (9821754), Nicorandil (47528), Pramipexole (119570), 2,4-dimethoxy-6-methylbenzene-1,3-diol (87294508), Hordenine (68313), Ergothioneine (5351619), Dencichine (440259), and BML-111 (10899465).

## Protein structures validation

The selected protein structures were validated using PROCHECK of SAVES v6.1 (https://saves.mbi.ucla.edu/) [17]. The protein structure was added. The PROCHECK was run. The main Ramachandran plot was used to analyze the distribution of phi and psi of residues of ERK2, JNK2, p38, p65, and IκBα. Based on the main Ramachandran plot file, if > 90% residues were in the most favored regions, the protein structure was considered a good quality model.

## Molecular docking

The compound structures in Structure Data File (SDF) format were downloaded from Pub-Chem (https://pubchem.ncbi.nlm.nih.gov/) [18]. The energy of the compounds was minimized and the compound structures were transferred to PDB format using OpenBabel-3.1.1. The compound structures were saved in PDB, Partial Charge, and Torsional degrees of freedom (PDBQT) format using AutoDock Vina [19].

The protein structures of ERK2, JNK2, p38, and p65 in PDB format were downloaded from PDB. The protein structure of IκBα in PDB format was downloaded from AFDB. The protein structures were prepared using AutoDock Vina. The nonpolar hydrogens of the proteins were

merged. The hydrogens of the proteins were added. The water of the proteins was deleted. The protein structures were saved in PDBQT format. The entire protein was selected for molecular docking. Because p65 and IκBα were too large to fit into the grid box, the major portions from the left and right sides of each protein were used for molecular docking.

Molecular docking between 51 compounds and 5 proteins was performed using AutoDock Vina. The complex structures were saved in PDB format. Among 10 or 20 configurations of each compound-protein complex, the configuration with minimal binding energy was selected to represent the docking result.

## Selection of compounds for drug-likeness, pharmacokinetics, bioactivity, and toxicity analysis, and cytotoxicity and anti-osteoarthritis effect testing

For compounds docked with ERK2, JNK2, p38, p65, and IκBα, although the configuration with minimal binding energy was selected to represent the result of molecular docking, each compound still had five binding energies. Because the individual contributions of ERK2, JNK2, p38, p65, and IκBα in OA are unknown, the average weight method was used in this complex multi-target screening scenario. The first binding energy score formula was used to screen compounds competitively binding to ERK2, JNK2, p38, and p65 but not IκBα.

$$Score_1 = E_{ERK2} + E_{JNK2} + E_{p38} + 3\left(E_{p65} - E_{I"B-}\right)$$

In this formula, equal weight was assigned to the MAPK and NFκB pathways and equal weight was assigned to the proteins in each pathway. The compounds were ranked from lowest to highest based on their binding energy scores.

Because in molecular docking, a lower the binding energy indicates a more stable complex and a stronger binding affinity between the molecules, top 10 compounds ranked by score of binding energy were selected. Because many inflammations share similar molecular mechanisms and many compounds with anti-inflammatory effect can inhibit various types of inflammations, 6 compounds with anti-inflammatory effect were selected out of the top 10 compounds [20,21]. They were Rutaecarpine, Corilagin, Apigetrin, Protopine, 5-methoxyflavone, and 7,3',4'-trihydroxyisoflavone. However, Rutaecarpine was newly reported to show anti-osteoarthritis effect [22]. Although it partly demonstrated the effectiveness of our methods, Rutaecarpine was excluded. The compounds selected for drug-likeness, pharmacokinetics, bioactivity, and toxicity analysis, and cytotoxicity and anti-osteoarthritis effect testing were Corilagin, Apigetrin, Protopine, 5-methoxyflavone, and 7,3',4'-trihydroxyisoflavone.

Because a reviewer suggested exploring other weighting method, the second binding energy score formula was used to screen compounds competitively binding to ERK2, JNK2, p38, and p65 but not IκBα.

$$Score_2 = E_{ERK2} + E_{JNK2} + E_{p38} + E_{p65} - E_{I"B-}$$

In this formula, equal weight was assigned to the proteins in each pathway. The compounds were ranked from lowest to highest based on their binding energy scores.

## Drug-likeness, pharmacokinetics, bioactivity, and toxicity analysis

The drug-likeness, pharmacokinetics, bioactivity, and toxicity of Corilagin, Apigetrin, Protopine, 5-methoxyflavone, and 7,3',4'-trihydroxyisoflavone were analyzed. The compound structures were the same as the compound structures used in molecular docking.

The drug-likeness of the compounds was analyzed based on Lipinski's rule of 5: molecular weight < 500 g/mol, LogP < 5, hydrogen bond donor count < 5, and hydrogen bond acceptor count < 10 [23]. The molecular weight, LogP, hydrogen bond donor count, and hydrogen bond acceptor count were from PubChem.

The pharmacokinetics of the compounds were predicted using SwissADME (http://www.swissadme.ch/) [24]. The compound structure was added. The sketched structure was transferred to the input list of Simplified Molecular Input Line Entry System (SMILES). The gastrointestinal absorption, blood brain barrier permeant, and P-glycoprotein substrate were calculated.

The bioactivity of the compounds was scored using Molinspiration (https://www.molinspiration.com/) [25]. The compound structure was added. The G protein-coupled receptor ligand, ion channel modulator,kinase inhibitor, nuclear receptor ligand,protease inhibitor,and enzyme inhibitor were calculated. However, the Molinspiration bioactivity score calculations had been removed in December 2024.

The toxicity of the compounds was predicted using ProTox 3.0 (https://tox.charite.de/protox3/) [26]. The compound structure was added. The oral lethal dose 50 (mg/kg),hepatotoxicity, neurotoxicity,nephrotoxicity,respiratory toxicity, and cardiotoxicity were calculated.

## Chondrocyte extraction

C57BL/6 mice were euthanized by cervical dislocation. Articular cartilage from the femoral head, femoral condyle, and tibial plateau was harvested, cut into pieces and washed several times with phosphate-buffered saline (PBS). The cartilage pieces were digested in a water bath with 0.1% type II collagenase and 0.25% trypsin for 1.5 h. The mixture was neutralized and centrifuged. The supernatant was discarded. The residue was digested with 0.1% type II collagenase overnight. The composition was mixed several times and filtered through a 100μm cell filter to remove tissue. The residue was centrifuged. The supernatant was discarded.Complete chondrocyte culture medium was added to a centrifuge tube. After mixing, the chondrocytes were added to a culture flask.

## Chondrocyte identification

Because during the chondrocyte extraction, the tissue near the articular cartilage, including muscle, ligament, and bone, may have been mixed with the harvested tissue, the chondrocytes were identified using immunofluorescence.

Coverslips were placed in a 24-well plate. Generation 1 chondrocytes (20000 cells in 1 ml complete chondrocyte culture medium/well) were added to the 24-well plate. When the chondrocytes adhered to the coverslips, the culture medium was discarded, and the coverslips were washed with PBS. The chondrocytes were fixed with 4% paraformaldehyde at 4°C for 30 min. The coverslips were washed several times with PBS. The permeabilization and blocking buffer was composed of 0.5% Triton X-100 (45%), PBS (45%), and goat serum (10%). The permeabilization and blocking buffer (50 μl) was placed on a parafilm. The coverslip was placed cell-side down on the buffer for 2 h. Type II collagen (50 μl, 1:100 in PBS) was placed on a parafilm. The coverslip was placed cell-side down on the solution at 4°C overnight. Alexa Fluor 488 labeled goat anti-rabbit lgG (50 μl, 1:100 in PBS) was placed on a parafilm. The coverslip was placed cell-side down on the solution for 2 h. The coverslips were washed several times with PBS. The chondrocytes were stained with DAPI (1:1000 in PBS) for 5 min and washed several times with PBS. A drop of Fluoromount-G was placed on the coverslip. A coverslip was placed on it.

## Cytotoxicity testing

The cytotoxicity of Corilagin, Apigetrin, Protopine, 5-methoxyflavone, and 7,3',4'-trihydroxyisoflavone was tested using the Cell Counting Kit-8. Generation 1 chondrocytes (5000 cells in 90 μl DMEM/F12 complete culture medium/well) were added to 96-well plates. After 1 d, DMEM/F12 complete culture medium with and without the compounds (10 μl/well) was added to the 96-well plates. The compounds in the wells were Corilagin (0, 5,10, 20, 40, 80, and 160 μg/ml), Apigetrin (0, 0.78125, 1.5625, 3.125, 6.25, 12.5, 25, 50, and 100 μg/ml), Protopine (0, 7.8125, 15.625, 31.25, 62.5, 125, 250, 500, 1000, and 2000 μg/ml), 5-methoxyflavone (0, 0.625, 1.25, 2.5, 5, 10, and 20 μg/ml), and 7,3',4'-trihydroxyisoflavone (0, 1.5625, 3.125, 6.25, 12.5, 25, 50, 100, and 200 μg/ml). After 1 d, the supernatant was removed and Cell Counting Kit-8 reagent (110 μl/well) was added to the 96-well plates. The 96-well plates were placed at 37 °C for 3-8 h. The optical density at 450 nm in the wells was measured. According to the guidelines outlined in ISO 10993-5:2009, if the cell viability of a treatment group was ≥ 70% of the cell viability of the control group, the compound of the treatment group was considered noncytotoxic [27].

## Anti-osteoarthritis effect testing

The anti-osteoarthritis effect of Corilagin, Apigetrin, Protopine, 5-methoxyflavone, and 7,3',4'-trihydroxyisoflavone were tested using the NO Detection Kit. Generation 1 chondrocytes (50000 cells in 300 μl DMEM/F12 complete culture medium/well) were put into 24-well plates. After 1 d, DMEM/F12 complete culture medium with IL-1 β (100 μl/well) and with and without the compounds (100 μl/well) was added to the 24 well plates. The concentration of IL-1 β in wells was 10 ng/ml. The noncytotoxic concentrations of the compounds was selected based on the tested cytotoxicity. Because the cell viabilities of Corilagin of 80 and 160 μg/ml and 7,3',4'-trihydroxyisoflavone of 200μg/ml were < 70% of the control groups, they were considered cytotoxic to chondrocytes. Because the cell viabilities of the other treatment groups were ≥ 70% of the control groups, they were considered noncytotoxic to chondrocytes (Fig 12). The compounds in the wells were Corilagin (0, 3.125, 6.25, 12.5, 25, and 50 μg/ml), Apigetrin (0, 5, 10, 20, and 40 μg/ml), Protopine (0, 62.5, 125, 250, 500, and 1000 μg/ml), 5-methoxyflavone (0, 2.5, 5, 10, and 20 μg/ml), and 7,3',4'-trihydroxyisoflavone (0, 2.5, 5, 10, and 20 μg/ml). After 1 d, the supernatant was centrifuged at 1000 g for 10 min. Part of the supernatant in every centrifuge tube (400 μl/centrifuge tube) was collected. The NO concentration in the supernatant was tested according to the specification of the NO Detection Kit (S1 Specification). The optical density at 550 nm in the wells of microplate were measured.

## Statistical analysis

GraphPad Prism 9 was used for all statistical analyses. Unpaired t test was used to compare the means of two groups. The P value was two-tailed. A P value < 0.05 was considered statistically significant.

# Results

## Proteins and compounds for molecular docking

The proteins used for molecular docking were ERK2 (8AOJ), JNK2 (7N8T), p38 (6QDZ), p65 (1NFI), and IκBα (AF-P25963-F1-v4). The compounds used for molecular docking were Rutaecarpine (65752), Sesamolin (101746), Corilagin (73568), Apigetrin (5280704), Gamabufotalin (259803), Protopine (4970), 5-methoxyflavone (94525),

7,3',4'-trihydroxyisoflavone (5284648), Viaminate (6438483), KAN0438757 (71586631), Savinin (5281867), Maraviroc (3002977), 6-methoxyflavone (147157), Sargachromenol (10455044), Tryptanthrin (73549), Chryseriol (5280666), Mogrol (14525327), Nuciferine (10146), Dehydromiltirone (3082765), Rosmanol (13966122), W54011 (5311122), Oxocrebanine (3084713), Chrysoeriol (5280666), Matairesinol (119205), Fluorofenidone (11851183), Demethoxycurcumin (5469424), Eupatilin (5273755), Xanthenone (7020), Tetrahydropalmatine (72301), Harmine (5280953), Isosinensetin (632135), Xanthotoxol (65090), Cl-Amidine (24970878), 3,3',4,5'-tetramethoxy-trans-stilbene (5387294), Pteryxin (5281425), 6-methylcoumarin (7092), Ibudilast (3671), Fingolimod (107970), Xanthatin (5281511), Guanosine (135398635), Nardostachin (196699), Dehydrozingerone (5354238), Scopoletin (5280460), Bicyclol (9821754), Nicorandil (47528), Pramipexole (119570), 2,4-dimethoxy-6-methylbenzene-1,3-diol (87294508), Hordenine (68313), Ergothioneine (5351619), Denichine (440259), and BML-111 (10899465) (S2 Structure).

## Protein validation

The percentage of residues in the most favored region for ERK2, JNK2, p38, p65, and IκBα were 90.3%, 92.2%, 90.8%, 79.0%, and 80.9%, respectively (Fig 3).

## Molecular docking and compounds for drug-likeness, pharmacokinetics, bioactivity, and toxicity analysis, and cytotoxicity and anti-osteoarthritis effect testing

The prepared proteins and compounds are shown in S3 Structure. The configurations and complexes are shown in S4 Structure.

A total of 22 compounds showed a binding energy of < -7 kcal/mol when docked with ERK2, JNK2, p38, and p65, and a binding energy of ≥ -7 kcal/mol when docked with IκBα. The top 10 compounds ranked based on the first binding energy score formula were Rutaecarpine, Sesamolin, Corilagin, Apigetrin, Gamabufotalin, Protopine, 5-methoxyflavone, 7,3',4'-trihydroxyisoflavone, Viaminate, and KAN0438757 (Fig 4, S5 Table).

$$Score_1 = E_{ERK2} + E_{JNK2} + E_{p38} + 3\left(E_{p65} - E_{I''B-}\right)$$

Compounds selected for drug-likeness, pharmacokinetics, bioactivity, and toxicity analysis, and cytotoxicity and anti-osteoarthritis effect testing are Corilagin, Apigetrin, Protopine, 5-methoxyflavone, and 7,3',4'-trihydroxyisoflavone. The positional relationships between Corilagin, Apigetrin, Protopine, 5-methoxyflavone, and 7,3',4'-trihydroxyisoflavone and ERK2, JNK2, p38, p65, and IκBαand the interactions between these compounds and residues of these proteins in molecular dockings are shown in Figs 5–10.

## Drug-likeness, pharmacokinetics, bioactivity, and toxicity

Drug-likeness analysis based on Lipinski's rule of 5, pharmacokinetic prediction, Molinspiration bioactivity scores, and toxicity prediction for Corilagin, Apigetrin, Protopine, 5-methoxyflavone, and 7,3',4'-trihydroxyisoflavone are shown in Tables 1–4.

## Chondrocyte identification

In immunofluorescence images, green fluorescence indicated type II collagen, and blue fluorescence indicated nucleus. The two colors co-localized well within the same cells. The extracted cells were identified as chondrocytes (Fig 11).

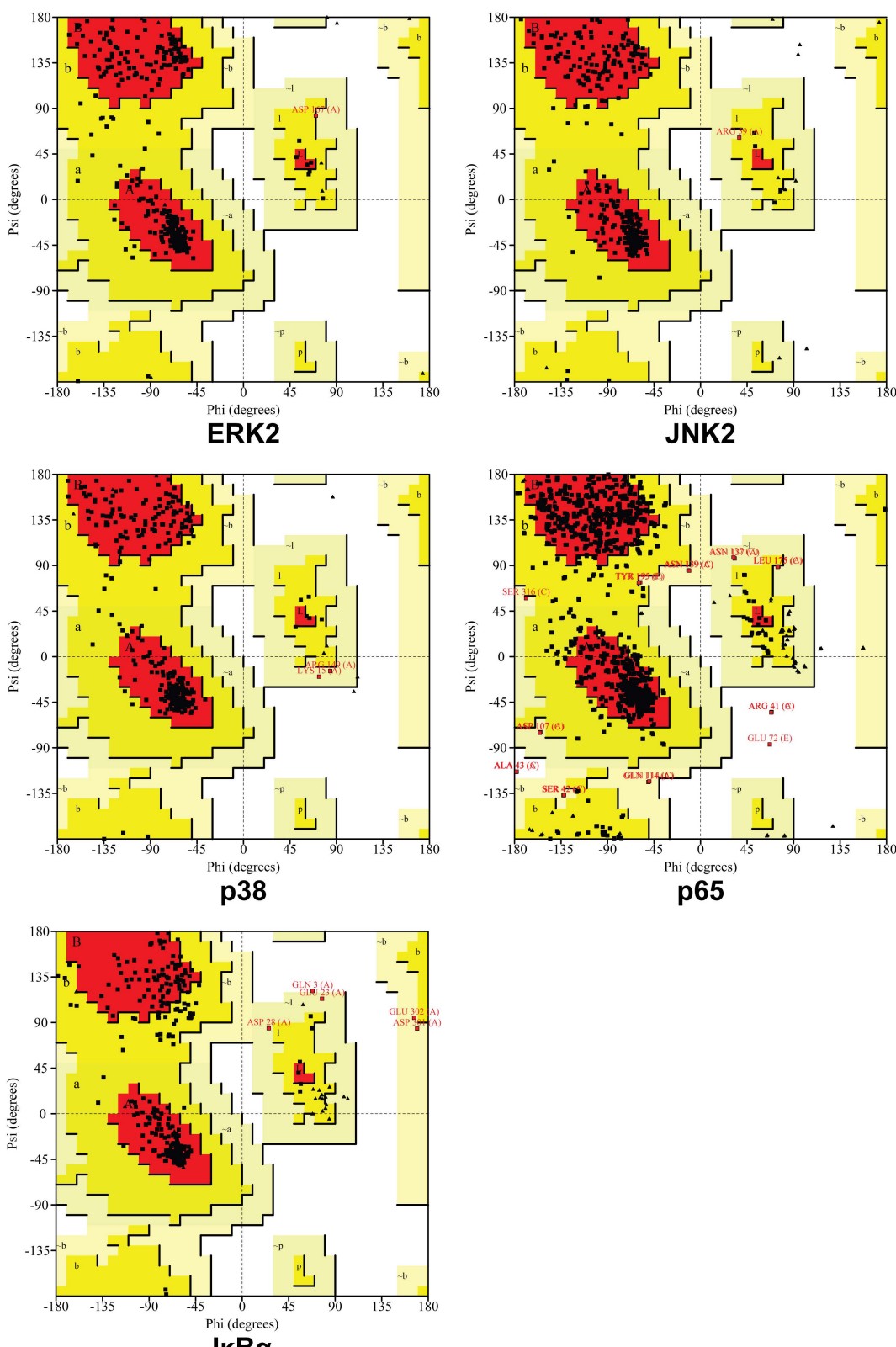

**Fig 3. Distribution of phi and psi of residues.** A, B, and L indicate most favored regions. a, b, l, and p indicate additional allowed regions. ~ a, ~ b, ~ l, and ~ p indicate generously allowed regions. Triangles indicate glycine residues.

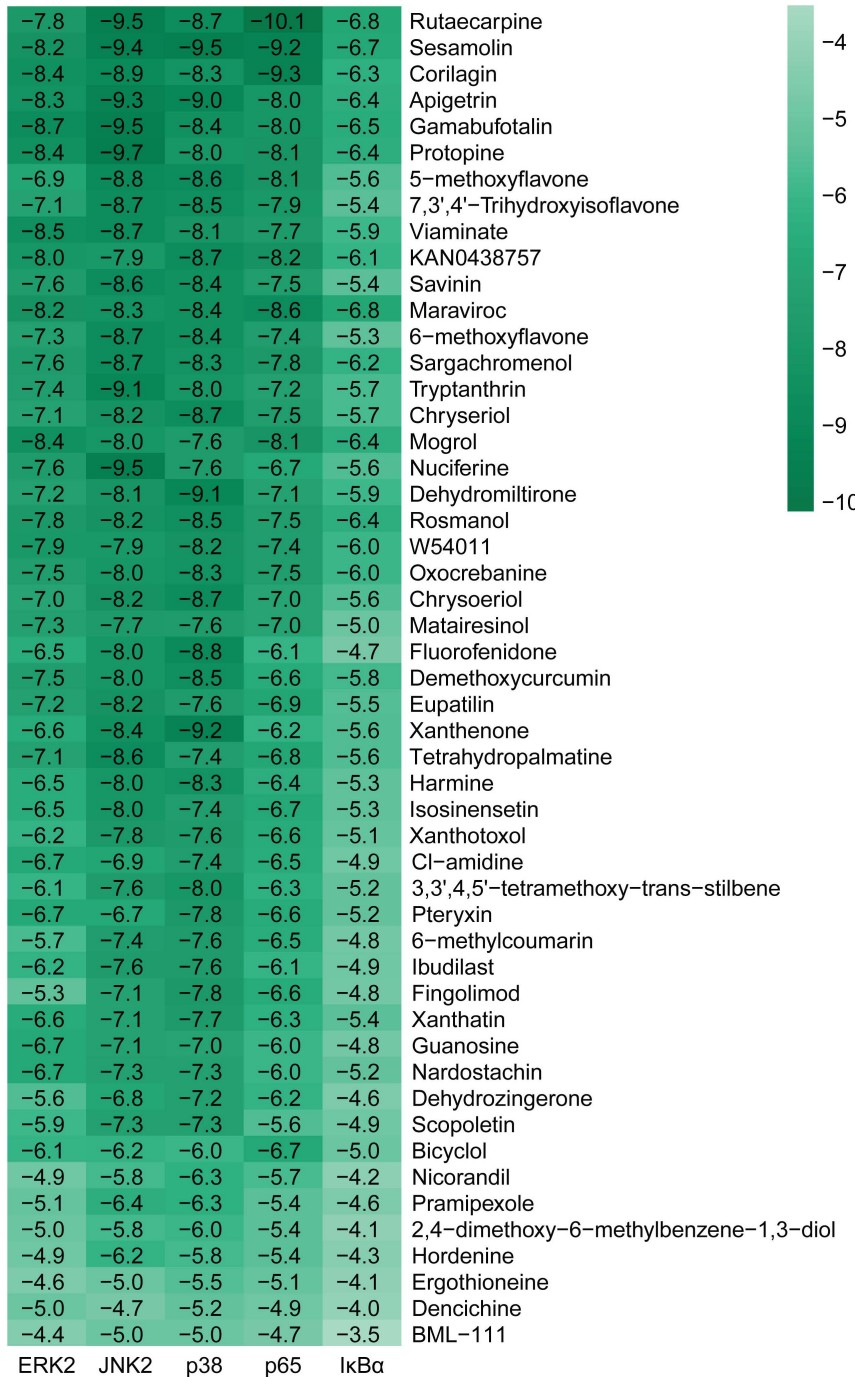

Fig 4. Bonding energies (kcal/mol) of molecular dockings between compounds and proteins. Compounds were ranked from lowest to highest based on the first binding energy score formula.

## Cytotoxicity

Cell viabilities of Corilagin of 80 and 160 μg/ml and 7,3',4'-trihydroxyisoflavone of 200μg/ml were < 70% of the cell viabilities of the control groups. The cell viabilities of the other treatment group were ≥ 70% of the cell viabilities of the control groups (Fig 12, S7 Table).

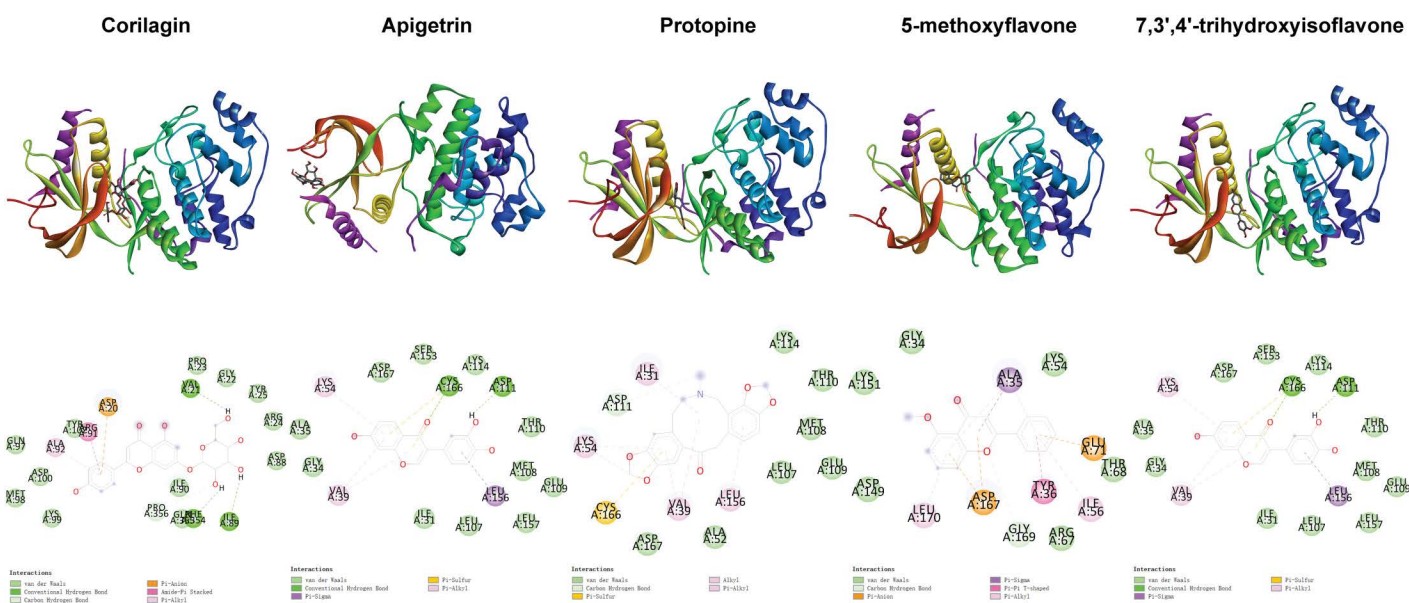

**Fig 5. Positional relationships between compounds and ERK2 and interactions between compounds and residues.**

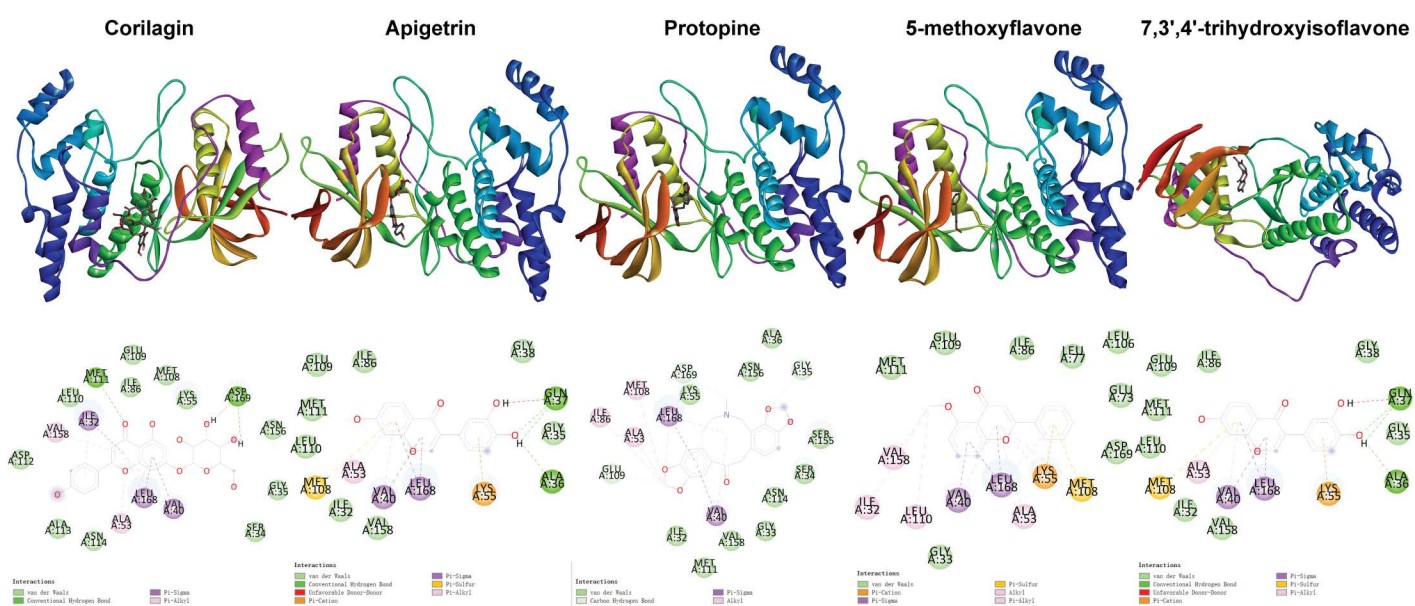

**Fig 6. Positional relationships between compounds and JNK2 and interactions between compounds and residues.**

## Anti-osteoarthritis effect

Corilagin of 3.125, 6.25, 12.5, and 25 µg/ml induced the release of NO in chondrocytes. Protopine of 62.5, 125, 250, 500, and 1000 µg/ml, 5-methoxyflavone of 2.5, 5, 10, and 20 µg/ml, and 7,3',4'-trihydroxyisoflavone of 5, 10, and 20 µg/ml inhibited the release of NO in chondrocytes. In other cases, the compounds did not affect the release of NO in chondrocytes (Fig 13, S8 Table).

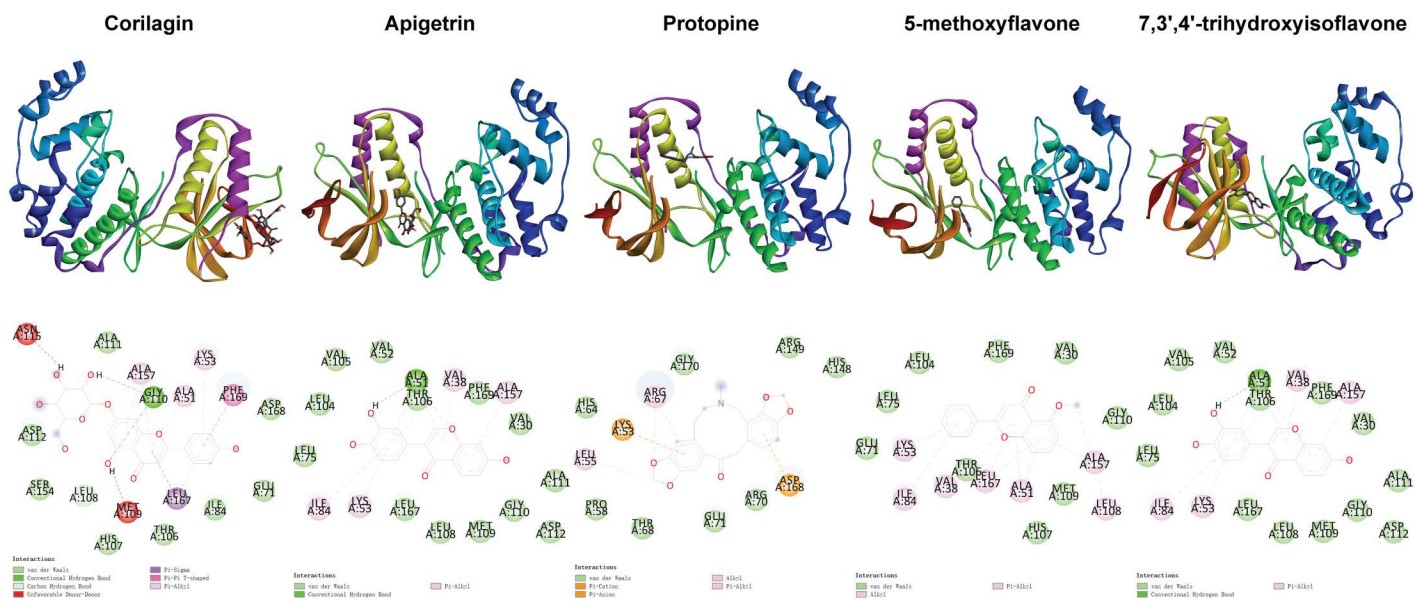

**Fig 7. Positional relationships between compounds and p38 and interactions between compounds and residues.**

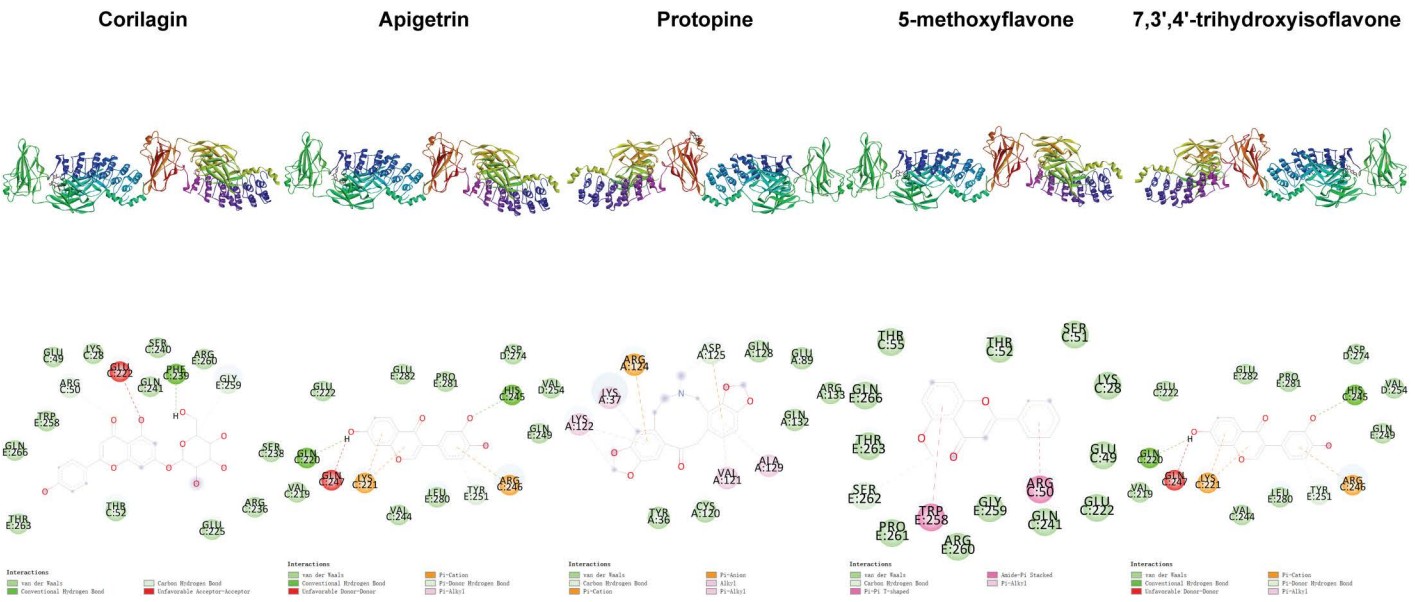

**Fig 8. Positional relationships between compounds and p65 and interactions between compounds and residues.**

## Discussion

This study found that Protopine, 5-methoxyflavone, and 7,3',4'-trihydroxyisoflavone show an anti-osteoarthritis effect (Fig 13). Protopine is found in *Corydalis yanhusuo* W.T. Wang, a famous herb of Traditional Chinese Medicine. Protopine shows wide pharmacological effects such as anti-inflammatory, anti-platelet aggregation, anti-cancer, analgesic, vasodilatory, anticholinesterase, anti-addictive, anticonvulsant, antipathogenic, antioxidant, hepatoprotective, neuroprotective, cytotoxic, and anti-proliferative effects [28]. 5-methoxyflavone is found in *Vochysia divergens*

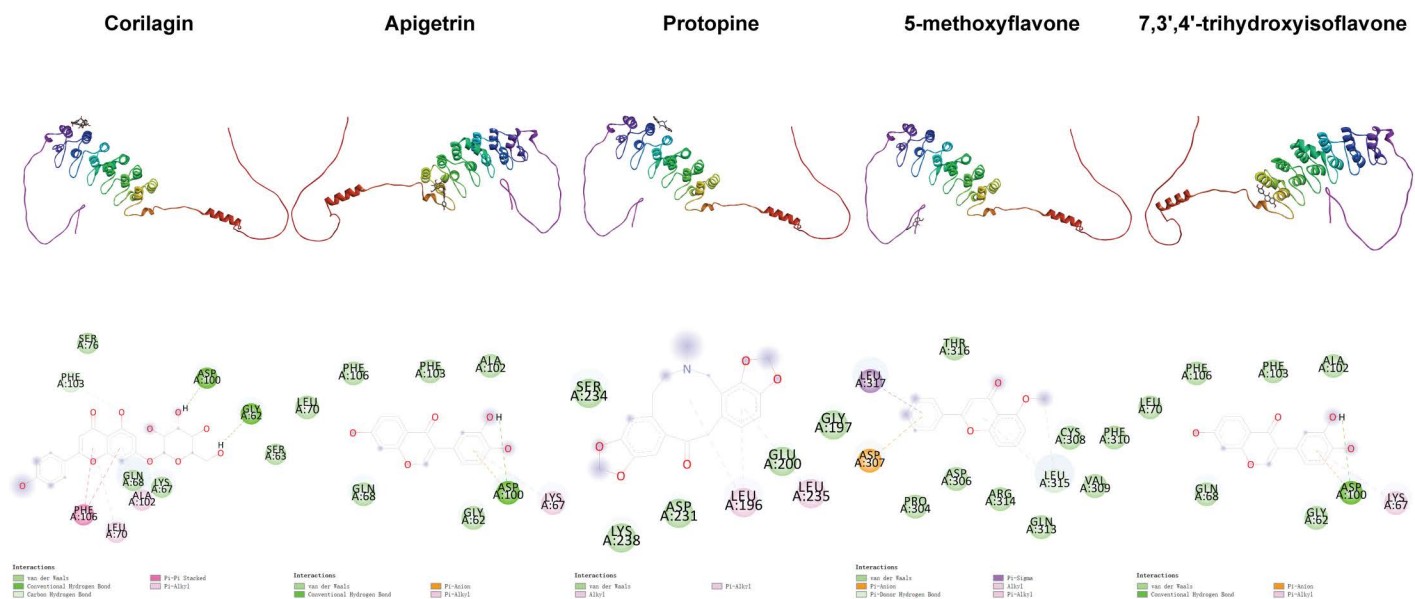

**Fig 9. Positional relationships between compounds and IκBα and interactions between compounds and residues.**

Pohl [29]. 5-methoxyflavone alleviates LPS-mediated lung injury and shows a neuroprotective effect [30,31]. 7,3',4'-trihydroxyisoflavone is a metabolite of daidzein, a representative isoflavone found in soybean, *Glycine max* (L.) Merr [32]. 7,3',4'-trihydroxyisoflavone shows anti-inflammatory and anti-cancer effects [32–34]. Future studies can test their anti-osteoarthritis effect in animal models, explore molecular mechanisms, and develop novel compounds based on the identified binding sites. Solid dispersion, nanotechnology, and salt formation can be used to improve their solubility and bioavailability [35–37].

This study used the average weight method to rank compounds in this complex multi-target screening scenario. If the importance of each protein in multi-target screening can be quantified or ranked, assigning weights to the proteins based on their importance will be appropriate. Similarly, some studies used the average weight method [38,39]. However, many studies did not rank compounds based on docking with multiple targets [38,40–43]. Additionally, in this study, among the binding energies of configurations of complex between a compound and a protein, the minimal binding energy was selected. Because minimal binding energy indicates the most stable binding model, selecting the minimal binding energy after molecular docking is very common. However, averaging binding energies can also be considered when evaluating comprehensive binding models [44].

Among the molecular dockings between Corilagin, Apigetrin, Protopine, 5-methoxyflavone, and 7,3',4'-trihydroxyisoflavone and ERK2, JNK2, p38, p65, and IκBα, the positional relationships show some interesting patterns. ERK2, JNK2, and p38 exhibited similar configurations, and the compounds commonly bound to the binding pockets composed of residues represented by red, yellow, and green. These binding pockets are important in drug discovery and drug design for diseases related to ERK2, JNK2, and p38. For p65 and IκBα, no single significant binding pocket was found (Figs 5-9).

The interactions also show some interesting patterns. The number of interactions between the compounds and residues depended on both the compounds and proteins. The interactions between the compounds and ERK2, JNK2, p38, and p65 are more numerous compared with the interactions between the compounds and IκBα. This pattern is consistent with the

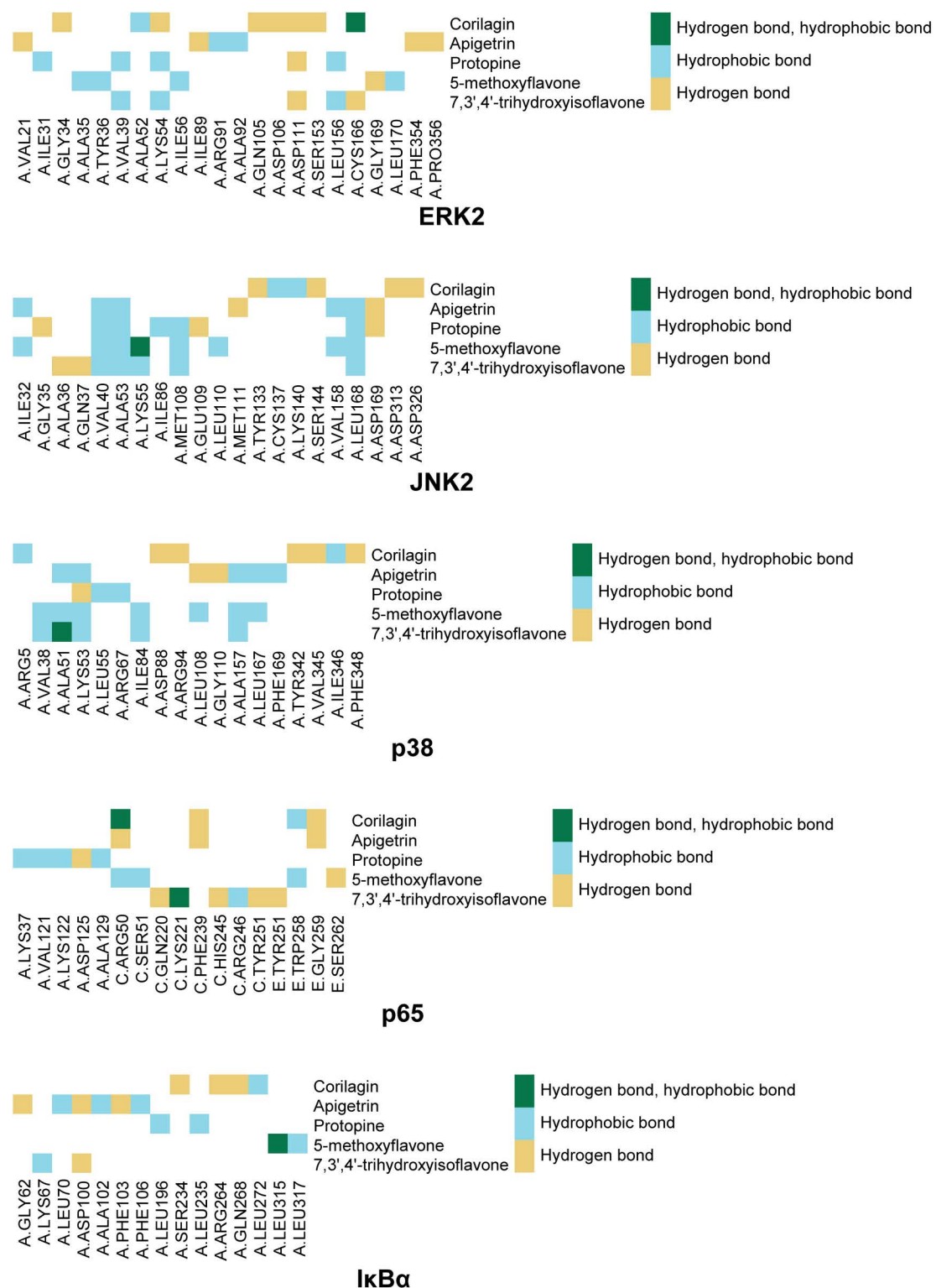

**Fig 10. Interactions between compounds and residues.** The top 10 compounds ranked based on the second binding energy score formula are Sesamolin, Rutaecarpine, Corilagin, Apigetrin, Gamabufotalin, Protopine, Viaminate, 5-methoxyflavone, 7,3',4'-Trihydroxyisoflavone, and Savinin (S6 Table).

binding energy scores formula used to screen compounds competitively binding to ERK2,

**Table 1. Drug-likeness analysis of selected compounds based on Lipinski's rule of 5.**

| Compound | Molecular weight (g/mol) | XLogP3 | Hydrogen bond donor count | Hydrogen bond acceptor count | Number of Lipinski's rule of 5 violations |
|---|---|---|---|---|---|
| Corilagin | 634.5 | 0.1 | 11 | 18 | 3 |
| Apigetrin | 432.4 | −0.1 | 6 | 10 | 1 |
| Protopine | 353.4 | 2.8 | 0 | 6 | 0 |
| 5-methoxyflavone | 252.26 | 3.5 | 0 | 3 | 0 |
| 7,3',4'-trihydroxyisoflavone | 270.24 | 2.1 | 3 | 5 | 0 |

**Table 2. Pharmacokinetic prediction of selected compounds.**

| Compound | Gastrointestinal absorption | Blood brain barrier permeant | P-glycoprotein substrate |
|---|---|---|---|
| Corilagin | Low | No | Yes |
| Apigetrin | Low | No | Yes |
| Protopine | High | Yes | Yes |
| 5-methoxyflavone | High | Yes | Yes |
| 7,3',4'-trihydroxyisoflavone | High | Yes | Yes |

**Table 3. Molinspiration bioactivity score of selected compounds.**

| Compound | G protein-coupled receptor ligand | Ion channel modulator | Kinase inhibitor | Nuclear receptor ligand | Protease inhibitor | Enzyme inhibitor |
|---|---|---|---|---|---|---|
| Corilagin | −0.11 | −0.71 | −0.45 | −0.44 | −0.03 | −0.15 |
| Apigetrin | 0.10 | −0.01 | 0.14 | 0.31 | 0.02 | 0.43 |
| Protopine | 0.18 | −0.04 | −0.26 | −0.23 | −0.03 | 0.04 |
| 5-methoxyflavone | −0.18 | −0.18 | 0.04 | 0.03 | −0.37 | 0.07 |
| 7,3',4'-trihydroxyisoflavone | −0.24 | −0.60 | −0.10 | 0.11 | −0.77 | 0.06 |

**Table 4. Toxicity prediction of selected compounds.**

| Compound | Oral lethal dose 50 (mg/kg) | Hepatotoxicity | Neurotoxicity | Nephrotoxicity | Respiratory toxicity | Cardiotoxicity |
|---|---|---|---|---|---|---|
| Corilagin | 2260 | − | − | + | + | − |
| Apigetrin | 5000 | − | − | + | + | + |
| Protopine | 940 | − | + | − | + | − |
| 5-methoxyflavone | 4000 | − | − | + | + | − |
| 7,3',4'-trihydroxyisoflavone | 2500 | − | − | + | + | − |

"+" Indicates active. "−" Indicates inactive.

JNK2, p38, and p65 but not IκBα. The compounds ranked from largest to smallest based on the number of interactions between the compounds and the residues are Corilagin, Apigetrin, Protopine, 5-methoxyflavone, and 7,3',4'-trihydroxyisoflavone. This pattern is consistent with the ranking of compounds from lowest to highest based on their binding energy scores. The number of hydrogen bonds between Corilagin and the residues is more numerous compared with the number of hydrophobic bonds. The number of hydrophobic bonds between Protopine and 5-methoxyflavone and the residues is more numerous compared with the number of hydrogen bond. Additionally, for ERK2, JNK2, and p38, among the interactions with the

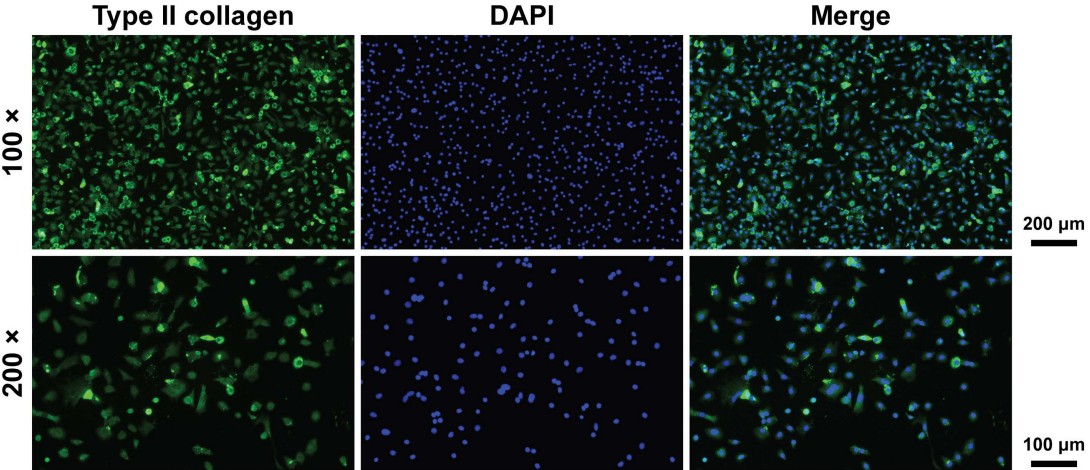

**Fig 11. Type II collagen and DAPI immunofluorescence staining images of mouse chondrocytes.**

compounds, the residues interacted with the compounds are similar (Figs 10). This similarity in the residues of the molecular dockings is consistent with the similarities in many studies [41,44,45].

In the drug-likeness analysis based on Lipinski's rule of 5, Protopine, 5-methoxyflavone, and 7,3',4'-trihydroxyisoflavone showed good drug-likeness because they violated no rules in Lipinski's rule of 5 (Table 1). In the pharmacokinetic prediction, Protopine, 5-methoxyflavone, and 7,3',4'-trihydroxyisoflavone were predicted to have good pharmacokinetic properties (Table 2). Similarly, Protopine, 5-methoxyflavone, and 7,3',4'-trihydroxyisoflavone showed an anti-osteoarthritis effect. Molecular weight $< 500\,\mathrm{g/mol}$, hydrogen bond donor count $< 5$, and hydrogen bond acceptor count $< 10$ suggest the compounds are easier to penetrate cell membranes and be absorbed. LogP $< 5$ suggests the compounds are easier to dissolve in water. These properties enable compounds to more easily penetrate the gastrointestinal barrier and the blood brain barrier. In cell experiments, compounds with these properties dissolve in the cell medium, penetrate cell membranes, reach target proteins, and exert pharmacological effects more easily. The anti-osteoarthritis effect of the compounds in this study is consistent with this pattern.

In the Molinspiration bioactivity score, Apigetrin might serve as a G protein-coupled receptor ligand, a nuclear receptor ligand, and an enzyme inhibitor. Protopine might serve as a G protein-coupled receptor ligand and an enzyme inhibitor. 5-methoxyflavone and 7,3',4'-trihydroxyisoflavone might serve as enzyme inhibitors (Table 3) [25]. Because ERK2, JNK2, and p38 are enzymes, compounds serving as enzyme inhibitors are more likely to inhibit ERK2, JNK2, and p38 and exert anti-osteoarthritis effect. The anti-osteoarthritis effect of the compounds in this study is partially consistent with this pattern.

In the toxicity prediction, Protopine was harmful if swallowed. Corilagin, 7,3',4'-trihydroxyisoflavone, 5-methoxyflavone, and Apigetrin may be harmful if swallowed [26]. Protopine showed significantly lower cytotoxicity compared with Corilagin, 7,3',4'-trihydroxyisoflavone, 5-methoxyflavone, and Apigetrin (Table 4). Because oral toxicity is associated with cytotoxicity to some extent, compounds predicted to have lower oral toxicity may have lower cytotoxicity [46]. The cytotoxicity of the compounds in this study is consistent with this pattern.

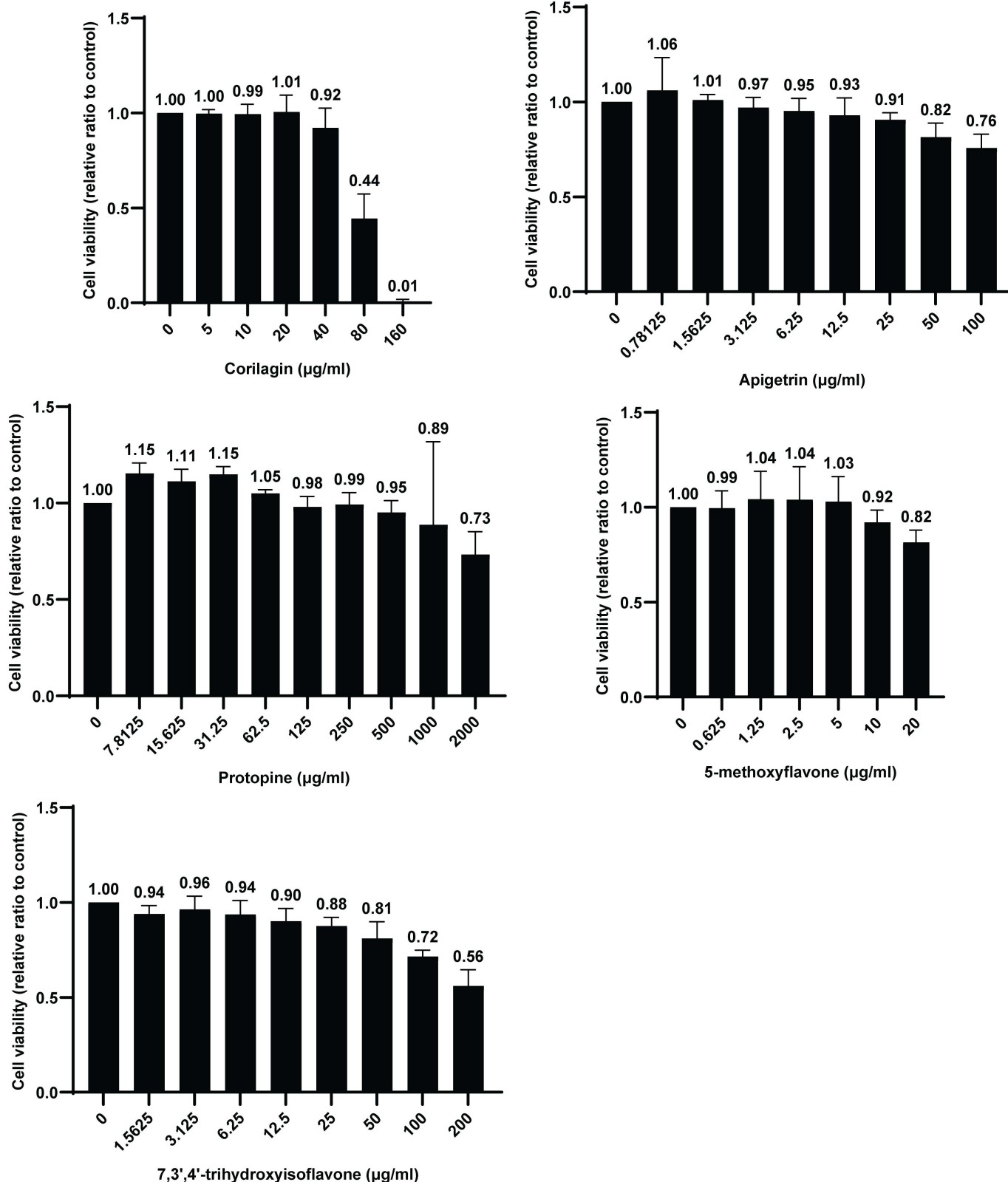

**Fig 12. Cell viability of mouse chondrocytes at 24h after treatment with compounds.**

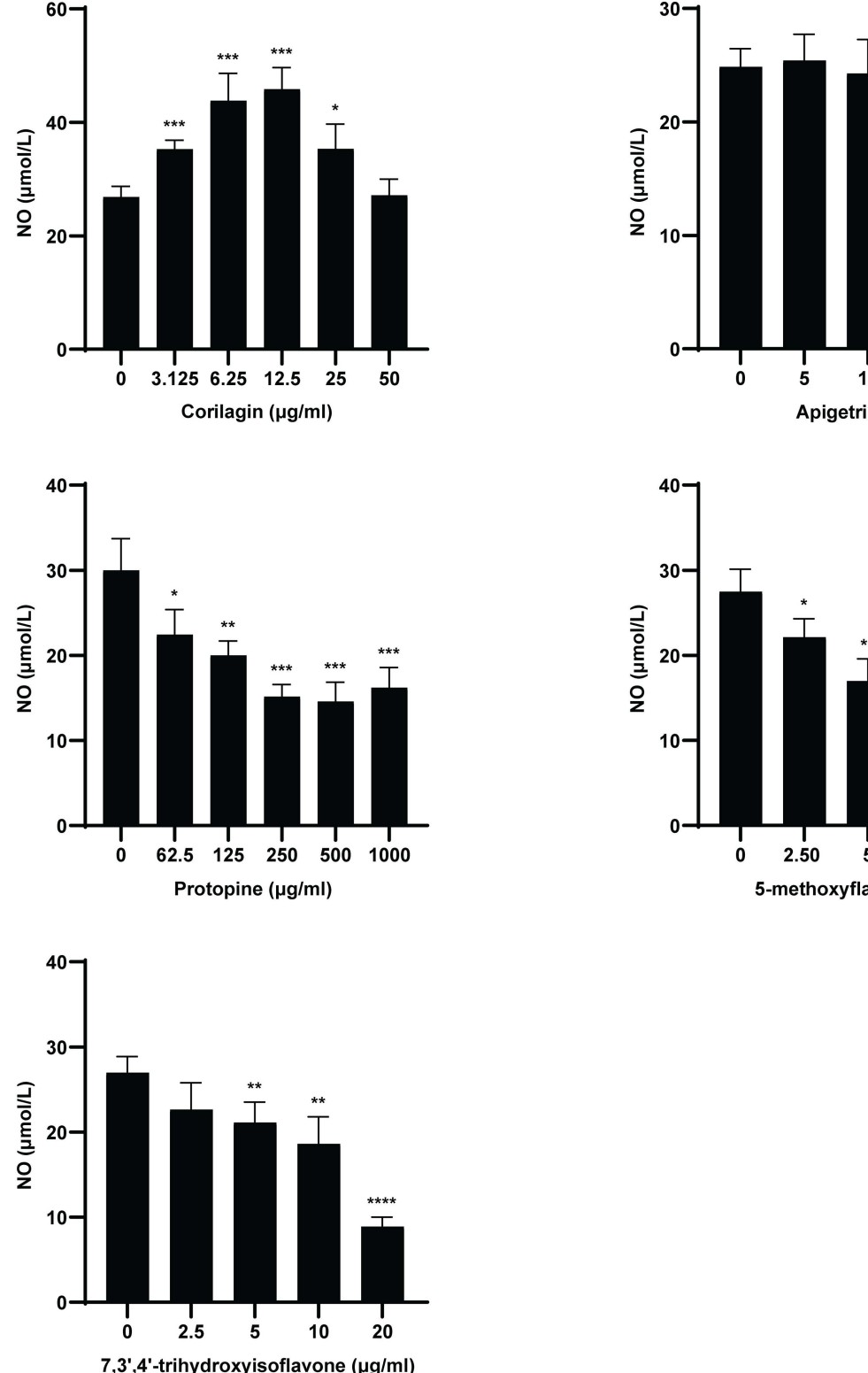

**Fig 13. NO concentration in the mouse chondrocyte culture medium at 24 h after treatment with compounds.** "*" indicates P < 0.05. "**" indicates P < 0.01. "***" indicates P < 0.001. "****" indicates P < 0.0001.

The first limitation of this study is the limited number of compounds used for molecular docking. These compounds inhibit MAPK and NFκB pathways but have not been used to treat osteoarthritis in studies published between 2021 and 2023. Nevertheless, 51 compounds and 255 molecular docking matches exceed the matches used in many studies using molecular docking to screen compounds with therapeutic potential. In the future, if the cost of molecular docking decreases significantly without losing its accuracy, screening potential anti-osteoarthritis compounds using molecular docking with a larger number of compounds can be feasible.

The second limitation of this study is the potential off-target effects. Although MAPK and NFκB pathways play important roles in the onset and progression of osteoarthritis, other pathways such as Wnt/β-catenin, focal adhesion, HIFs, and TGFβ/BMP are also significant [8]. Because of the complexity of the molecular interactions in osteoarthritis, avoiding the off-target effects in the osteoarthritis field is challenging. Although this study using multi-target screening rather than single-target screening helps reduce off-target effects, using more targets in molecular docking and protein microarray to screen anti-osteoarthritis compounds will be more useful. Additionally, protein microarray, immunoprecipitation, X-ray crystallography, and gene knockout models can be used to evaluate the interaction between compounds and proteins [47–50].

The third limitation of this study is that not all protein structures used for molecular docking are ideal. According to the main Ramachandran plot file of PROCHECK of SAVES v6.1, if > 90% residues are in the most favored regions, the protein structure is considered a good quality model. The percentage of residues in the most favored regions of p38 and p65 are only 79.0% and 80.9%, respectively. However, because the residues in generously allowed regions and disallowed regions of p38 and p65 are completely inconsistent with the residues interacting with the compounds in molecular docking, and the residues in the disallowed regions are rare, the molecular docking remains an appropriate tool for this screening. Additionally, the anti-osteoarthritis effect based on chondrocytes also validated the effectiveness of molecular docking in screening. In the future studies, molecular dynamics stimulation can be used to improve protein structures.

The fourth limitation of this study is using human protein structures for molecular docking while using mouse chondrocytes for validating anti-osteoarthritis effect of selected compounds. In our department, the Department of Sports Medicine, harvesting numerous human articular cartilages in joint replacement is very convenient and cost-saving. However, the severity of osteoarthritis and age may significantly influence chondrocyte properties. Although sacrificing mice, normal articular cartilage from the mice of the same age is more useful for consistently controlling irrelevant variables. Experiments based on the same species make the results more comparable. For clinical medicine rather than veterinary medicine, studies based on human tissues are more useful for clinical translation. Nevertheless, MAPK and NFκB pathways are highly conserved across different species [15,16]. Many studies on osteoarthritis used both human and mouse tissue [11,12]. Therefore, using both human protein structures and mouse chondrocytes in this study is valid.

## Conclusions

Molecular docking based on MAPK and NFκB pathways can be used to screen potential anti-osteoarthritis compounds, offering a pathway-based perspective for drug discovery. For the selected compounds, the theoretical drug-likeness, pharmacokinetics, bioactivity, and toxicity are largely consistent with the empirical cytotoxicity and anti-osteoarthritis effect, indicating the effectiveness of computational experiments. Additionally, Protopine,

5-methoxyflavone, and 7,3',4'-trihydroxyisoflavone showed strong anti-osteoarthritis potential and can be considered for future studies to test their anti-osteoarthritis effect in animal models, explore molecular mechanisms, and improve their solubility.

## Supporting information

**S1 Specification. Nitric Oxide Detection Kit.**
(PDF)

**S2 Structure. Proteins and compounds.**
(ZIP)

**S3 Structure. Prepared proteins and compounds.**
(ZIP)

**S4 Structure. Configurations and complexes.**
(ZIP)

**S5 Table. Compound ranked based on the first score formula.**
(XLSX)

**S6 Table. Compound ranked based on the second score formula.**
(XLSX)

**S7 Table. Cell viability.**
(XLSX)

**S8 Table. Nitric oxide concentration.**
(XLSX)

## Author contributions

**Conceptualization:** Tian-Wang Zhu.

**Funding acquisition:** Rui-Xin Li.

**Resources:** Yu Zheng.

**Writing – original draft:** Tian-Wang Zhu.

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
