## [Decision Letter · Decision Letter 0]

26 Nov 2024

PONE-D-24-47350Screening potential antiosteoarthritis compounds using molecular docking based on MAPK and NFκB pathways and effect verificationPLOS ONE

Dear Dr. Li,

Thank you for submitting your manuscript to PLOS ONE. After careful consideration, we feel that it has merit but does not fully meet PLOS ONE’s publication criteria as it currently stands. Therefore, we invite you to submit a revised version of the manuscript that addresses the points raised during the review process.

We look forward to receiving your revised manuscript.

Kind regards,

Hongxun Tao

Academic Editor

PLOS ONE

Journal Requirements: When submitting your revision, we need you to address these additional requirements. 1. Please ensure that your manuscript meets PLOS ONE's style requirements, including those for file naming. The PLOS ONE style templates can be found at https://journals.plos.org/plosone/s/file?id=wjVg/PLOSOne_formatting_sample_main_body.pdf and https://journals.plos.org/plosone/s/file?id=ba62/PLOSOne_formatting_sample_title_authors_affiliations.pdf

Reviewers' comments:

Reviewer's Responses to Questions

**Comments to the Author**

1. Is the manuscript technically sound, and do the data support the conclusions?

Reviewer #1: Yes

Reviewer #2: Yes

Reviewer #3: Yes

Reviewer #4: No

2. Has the statistical analysis been performed appropriately and rigorously? 

Reviewer #1: No

Reviewer #2: Yes

Reviewer #3: Yes

Reviewer #4: Yes

3. Have the authors made all data underlying the findings in their manuscript fully available?

Reviewer #1: No

Reviewer #2: Yes

Reviewer #3: No

Reviewer #4: Yes

4. Is the manuscript presented in an intelligible fashion and written in standard English?

Reviewer #1: Yes

Reviewer #2: Yes

Reviewer #3: No

Reviewer #4: No

5. Review Comments to the Author

Reviewer #1: I do have the following comments to improve the quality of the manuscript.

a. Protopine, 5-methoxyflavone, and 7,3',4'-trihydroxyisoflavone, which you mentioned, should also be subjected to 500-1000 nanosecond molecular dynamics simulation to investigate the stability of the interaction of the drug with the target protein using rmsd and hydrogen bond formation studies.

b. A list of catalytic site residues interacting with drugs and binding site pocket residues must be highlighted for each docking study.

c. A Github should be maintained to put all codes and input output files related to docking and molecular dynamics simulations.

d. Figure numbers 4-8 are not clear, especially the diagrams at the bottom area; please replace these diagrams with high-resolution images.

e. Ligplot+ based ligand protein interacrion could also be used for better clarity.

Reviewer #2: Comments for PONE-D-24-47350

I have gone through the assigned manuscript by Zhu et al., entitled “Screening potential antiosteoarthritis compounds using molecular docking based on MAPK and NFκB pathways and effect verification”, and found that it is competently well-written and the manuscript contains sufficient data but it should meet the following comments:

1-The abstract should provide a concise summary of including key findings of the study.

2-Authors suggest to add more recent citations with relevant studies (2023-2024).

3-The authors have to compare their results with literature data and improve this section completely.

4- Material and methods should include more details

5- Why did the author choose the different proteins for docking studies? Give your appropriate reasons properly.

6- How do authors validate the protein? Add Ramachandran plot.

7- Enrich the discussion section by referencing other relevant studies that would be nice.

8- I recommend adding the following five articles to the manuscript as citations.

https://doi.org/10.1016/j.chphi.2024.100753

https://doi.org/10.3390/molecules29143366

10.2174/0115734064304100240511112619

https://doi.org/10.1080/10286020.2024.2343821

https://doi.org/10.1016/j.jsps.2024.102062

9- References are not properly cited and should follow the journal style. The References section has to be revised after all corrections.

10- I suggest improving the English language.

My Opinion: Major Revision.

Reviewer #3: This study highlights the prevalence and impact of osteoarthritis, the limitations of current treatments, and the need for non-surgical alternatives. The focus on the MAPK and NFκB pathways as potential therapeutic targets is well-supported and leads logically to the study's objectives. The selection criteria for compounds and proteins are detailed, and the use of molecular docking is justified. The process of chondrocyte extraction and identification is also well-described.

The rationale for weighting p65 three times higher than other proteins is based on its central role in OA pathogenesis. However, the authors acknowledge the limitations of the equal weighting approach and suggest that future research could explore more sophisticated weighting schemes based on protein importance.

The authors acknowledge the limited number of compounds used in the molecular docking, which is a valid concern. The consistency between the empirical and theoretical results is emphasized, strengthening the study's conclusions.

The figures and tables are clear, well-organized, and effectively convey the study's data. The 3D configurations illustrating the positional relationships between the compounds and proteins are particularly helpful in understanding the molecular interactions.

While the study is well-executed overall, some areas could be strengthened:

Expand on the limitations of using human proteins for molecular docking while verifying effects on mouse chondrocytes. Although the authors acknowledge the species difference, a more detailed discussion of potential implications for translational relevance would be beneficial.

Address the potential for off-target effects. The compounds selected may interact with other proteins beyond those targeted in this study. Discussing the possibility of off-target effects and strategies for mitigating them would enhance the study's rigor.

While the chosen formula is reasonable, investigating other weighting approaches could provide additional insights and potentially identify different lead compounds.

Building upon the study's findings, outlining specific next steps for further research would enhance the paper's impact. This could involve testing the selected compounds in animal models of OA or exploring the development of novel compounds based on the identified binding identification.

English proofreading and editing need to be further improved

Reviewer #4: The manuscript titled "Screening potential anti-osteoarthritis compounds using molecular docking based on MAPK and NFκB pathways and effect verification" presents data with significant potential. However, the order of presentation is chaotic, making it difficult to follow. The English is good, but the disorderly arrangement of the manuscript hinders comprehension. If I were a reader unfamiliar with this research, I would find it even more challenging to grasp the content.

The authors have omitted crucial details about the preparation of the computational section, making it difficult for readers to replicate the research. A more comprehensive description of this section is necessary for the manuscript to be fully understood and for the research to be reproducible.

The conclusion is weak and needs more improvement.

It is clear that this manuscript requires a comprehensive rewrite before it can be considered for another submission. The current structure and presentation of the data make it difficult for readers to understand the research.

I do not recommend this manuscript for submission in its current state.

6. PLOS authors have the option to publish the peer review history of their article (what does this mean?). If published, this will include your full peer review and any attached files.

Reviewer #1: **Yes: **Swarna Kanchan

Reviewer #2: No

Reviewer #3: No

Reviewer #4: No

---

## [Author Response · Author response to Decision Letter 1]

17 Jan 2025

We sincerely appreciate the comments of the four reviewers. They have been very helpful to us.

Reviewer #1

Comment:

a. Protopine, 5-methoxyflavone, and 7,3',4'-trihydroxyisoflavone, which you mentioned, should also be subjected to 500-1000 nanosecond molecular dynamics simulation to investigate the stability of the interaction of the drug with the target protein using rmsd and hydrogen bond formation studies.

Response:

We are willing to use molecular dynamics simulation in future studies to explore the dynamic behaviors of molecules and evaluate the binding stability. However, the aim of this study is to screen potential antiosteoarthritis compounds and verify their effects. We believe that, although not perfect, the computational and biological experiments in this study have achieved the aim.

Comment:

b. A list of catalytic site residues interacting with drugs and binding site pocket residues must be highlighted for each docking study.

Response:

The residues interacting with the drugs have been listed in Figure 10.

Comment:

c. A Github should be maintained to put all codes and input output files related to docking and molecular dynamics simulations.

Response:

Considering that access to Github is restricted in some countries, the related files have been put in the Supporting Information section.

Comment:

d. Figure numbers 4-8 are not clear, especially the diagrams at the bottom area; please replace these diagrams with high-resolution images.

Response:

These diagrams have been replaced with high-resolution images.

Comment:

e. Ligplot+ based ligand protein interacrion could also be used for better clarity.

Response:

Thank you for sharing your experience.

Reviewer #2

Comment:

1-The abstract should provide a concise summary of including key findings of the study.

Response:

A concise summary including key findings of the study has been provided.

Comment:

2-Authors suggest to add more recent citations with relevant studies (2023-2024).

Response:

More recent citations with relevant studies (2023-2024) have been added.

Comment:

3-The authors have to compare their results with literature data and improve this section completely.

Response:

The results have been compared with the literature data.

Comment:

4- Material and methods should include more details

Response:

More details have been included in the Materials and methods section.

Comment:

5- Why did the author choose the different proteins for docking studies? Give your appropriate reasons properly.

Response:

The reasons for choosing the different proteins for docking studies were given in the Introduction section and have also been given in the Selection of proteins and compounds for molecular docking section.

Comment:

6- How do authors validate the protein? Add Ramachandran plot.

Response:

The Ramachandran plots have been added.

Comment:

7- Enrich the discussion section by referencing other relevant studies that would be nice.

Response:

The discussion section has been enriched by referencing other relevant studies.

Comment:

8- I recommend adding the following five articles to the manuscript as citations.

https://doi.org/10.1016/j.chphi.2024.100753

https://doi.org/10.3390/molecules29143366

10.2174/0115734064304100240511112619

https://doi.org/10.1080/10286020.2024.2343821

https://doi.org/10.1016/j.jsps.2024.102062

Response:

These articles have been added as citations.

Comment:

9- References are not properly cited and should follow the journal style. The References section has to be revised after all corrections.

Response:

The References section has been revised.

Comment:

10- I suggest improving the English language.

Response:

The English language has been improved.

Reviewer #3

Comment:

Expand on the limitations of using human proteins for molecular docking while verifying effects on mouse chondrocytes. Although the authors acknowledge the species difference, a more detailed discussion of potential implications for translational relevance would be beneficial.

Response:

The expand on the limitations have been added in the Discussion section.

Comment:

Address the potential for off-target effects. The compounds selected may interact with other proteins beyond those targeted in this study. Discussing the possibility of off-target effects and strategies for mitigating them would enhance the study's rigor.

Response:

The possibility of off-target effects and strategies for mitigating them have been discussed in the Discussion section.

Comment:

While the chosen formula is reasonable, investigating other weighting approaches could provide additional insights and potentially identify different lead compounds.

Response:

Another weighting approach has been added in the Selection of compounds for drug-likeness, pharmacokinetics, bioactivity, and toxicity analysis, and cytotoxicity and antiosteoarthritis effects testing section.

Comment:

Building upon the study's findings, outlining specific next steps for further research would enhance the paper's impact. This could involve testing the selected compounds in animal models of OA or exploring the development of novel compounds based on the identified binding identification.

Response:

Specific next steps for future research have been outlined.

Comment:

English proofreading and editing need to be further improved.

Response:

English proofreading and editing have been improved.

Reviewer #4

Comment:

The manuscript titled "Screening potential anti-osteoarthritis compounds using molecular docking based on MAPK and NFκB pathways and effect verification" presents data with significant potential. However, the order of presentation is chaotic, making it difficult to follow. The English is good, but the disorderly arrangement of the manuscript hinders comprehension. If I were a reader unfamiliar with this research, I would find it even more challenging to grasp the content.

Response:

The presentation has been modified. Many details have been added in the Materials and methods section to explain the order and content of the study design.

Comment:

The authors have omitted crucial details about the preparation of the computational section, making it difficult for readers to replicate the research. A more comprehensive description of this section is necessary for the manuscript to be fully understood and for the research to be reproducible.

Response:

The omitted crucial details about the preparation of the computational section have been added.

Comment:

The conclusion is weak and needs more improvement.

Response:

The conclusion has been improved.

Comment:

It is clear that this manuscript requires a comprehensive rewrite before it can be considered for another submission. The current structure and presentation of the data make it difficult for readers to understand the research.

Response:

This manuscript has been comprehensively rewritten.

---

## [Decision Letter · Decision Letter 1]

6 Feb 2025

Screening potential anti-osteoarthritis compounds using molecular docking based on MAPK and NFκB pathways and validating their anti-osteoarthritis effect

PONE-D-24-47350R1

Dear Dr. Li,

We’re pleased to inform you that your manuscript has been judged scientifically suitable for publication and will be formally accepted for publication once it meets all outstanding technical requirements.

Kind regards,

Hongxun Tao

Academic Editor

PLOS ONE

Additional Editor Comments (optional):

Reviewers' comments:

Reviewer's Responses to Questions

**Comments to the Author**

1. If the authors have adequately addressed your comments raised in a previous round of review and you feel that this manuscript is now acceptable for publication, you may indicate that here to bypass the “Comments to the Author” section, enter your conflict of interest statement in the “Confidential to Editor” section, and submit your "Accept" recommendation.

Reviewer #3: All comments have been addressed

Reviewer #4: All comments have been addressed

2. Is the manuscript technically sound, and do the data support the conclusions?

Reviewer #3: Yes

Reviewer #4: Yes

3. Has the statistical analysis been performed appropriately and rigorously? 

Reviewer #3: No

Reviewer #4: Yes

4. Have the authors made all data underlying the findings in their manuscript fully available?

Reviewer #3: No

Reviewer #4: Yes

5. Is the manuscript presented in an intelligible fashion and written in standard English?

Reviewer #3: (No Response)

Reviewer #4: Yes

6. Review Comments to the Author

Reviewer #3: The authors have made substantial revisions to the manuscript, "Screening potential anti-osteoarthritis compounds using molecular docking based on MAPK and NFκB pathways and validating their anti-osteoarthritis effect," and have addressed most of the major concerns raised in the previous review. The manuscript is now significantly improved in terms of clarity, methodology, and discussion. Only concern remaining is to improve the resolution of the figure from 5-11.

Reviewer #4: The author has done an excellent job in thoroughly revising the manuscript and addressing all of my concerns in a clear and comprehensive manner. The improvements made to the content demonstrate a significant level of effort and commitment to ensuring the quality and scientific rigor of the study. The revised manuscript is now well-structured, with clarified methodologies, strengthened discussions, and a more cohesive presentation of the results and conclusions.

I am particularly impressed with how the author incorporated the feedback, enhancing the clarity and depth of the sections that previously needed attention. These revisions have greatly improved the readability and impact of the manuscript, making it a valuable contribution to the field.

Given the thoughtful revisions and the high quality of the work, I have no hesitation in recommending this article for publication. Congratulations on a job well done, and I am confident this study will be of great interest to the scientific community.

7. PLOS authors have the option to publish the peer review history of their article (what does this mean?). If published, this will include your full peer review and any attached files.

Reviewer #3: No

Reviewer #4: No

---

## [Editor Report · Acceptance letter]

PONE-D-24-47350R1

PLOS ONE

Dear Dr. Li,

I'm pleased to inform you that your manuscript has been deemed suitable for publication in PLOS ONE. Congratulations! Your manuscript is now being handed over to our production team.

Kind regards,

on behalf of

Dr. Hongxun Tao

Academic Editor

PLOS ONE